# Sphingosine-1-phosphate Decreases Erythrocyte Dysfunction Induced by β-Amyloid

**DOI:** 10.3390/ijms25105184

**Published:** 2024-05-10

**Authors:** Francesco Misiti, Pierluigi Diotaiuti, Giovanni Enrico Lombardo, Ester Tellone

**Affiliations:** 1Human Sciences, Social and Health Department, University of Cassino and Lazio Meridionale, V. S. Angelo, Loc. Folcara, 03043 Cassino, Italy; p.diotaiuti@unicas.it; 2Department of Chemical, Biological, Pharmaceutical and Environmental Sciences, University of Messina, Viale Ferdinando Stagno d’Alcontres 31, 98166 Messina, Italy; giovanni.lombardo@unime.it (G.E.L.); ester.tellone@unime.it (E.T.)

**Keywords:** sphingosine-1-phosphate, erythrocyte, caspase-3, beta-amyloid, Alzheimer’s disease

## Abstract

Amyloid beta peptides (Aβ) have been identified as the main pathogenic agents in Alzheimer’s disease (AD). Soluble Aβ oligomers, rather than monomer or insoluble amyloid fibrils, show red blood cell (RBC) membrane-binding capacity and trigger several morphological and functional alterations in RBCs that can result in impaired oxygen transport and delivery. Since bioactive lipids have been recently proposed as potent protective agents against Aβ toxicity, we investigated the role of sphingosine-1-phosphate (S1P) in signaling pathways involved in the mechanism underlying ATP release in Ab-treated RBCs. In RBCs following different treatments, the ATP, 2,3 DPG and cAMP levels and caspase 3 activity were determined by spectrophotometric and immunoassay. S1P rescued the inhibition of ATP release from RBCs triggered by Ab, through a mechanism involving caspase-3 and restoring 2,3 DPG and cAMP levels within the cell. These findings reveal the molecular basis of S1P protection against Aβ in RBCs and suggest new therapeutic avenues in AD.

## 1. Introduction

Sphingosine-1-phosphate (S1P) is a potent lipid mediator that performs several roles [1]. Sphingosine kinase 1 (Sphk1) or sphingosine kinase 2 produce S1P from their precursor sphingosine; meanwhile, S1P phosphatase and S1P lyase (Sgpl) revert into sphingosine and 2-hexadecenal and phospho-ethanolamine, respectively [2]. Red blood cells (RBCs) uptake S1P [3,4,5], while S1P may also be produced within the cells through Sphk1 [2]. Since RBCs contain Sphk1 but no S1P-degrading enzymes [6], S1P is abundantly stored in RBCs [7], as well as in platelets [8] and the endothelium [9,10]. S1P performs several functions and regulates many cellular processes, including cell growth, proliferation, migration, and apoptosis [11,12,13,14]. In recent papers, S1P has been discussed concerning the RBC adaptation mechanism to SARS-CoV-2 infection [15]. In RBCs, S1P promotes deoxygenated haemoglobin (deoxyHb), which anchors to band 3, the most abundant membrane protein in RBCs, thereby increasing glycolysis flux, the 2,3-diphosphoglycerate (2,3 DPG) levels [16], and ATP release [17]. RBCs release ATP under reduced oxygen tensions and following deformation to modulate vasodilation [18]. The pathway underlying ATP release from RBCs involves several proteins, such as G proteins, adenylyl cyclase (AC), and cyclic AMP-dependent protein kinase A [18], which are a cystic fibrosis transmembrane conductance regulator and protein pannexin, respectively [19,20]. Alzheimer’s disease (AD) is a pathology characterized by senile plaques in several regions of the central nervous system (CNS), which are frequently correlated with areas of neurodegeneration [21]. Amyloid beta (Aβ) peptides, major protein components in the plaques, consist of 39–43 amino acid peptides originating from a more significant transmembrane protein, amyloid precursor protein (APP). Aβ neurotoxicity has been associated with peptide self-aggregation, which leads to the formation of amyloid-like fibrils [22] and eventually to neuronal cell death through apoptosis. However, recent studies have shown that soluble forms of Aβ exhibit stronger neurotoxicity, and in its monomeric form, Aβ may be responsible for the neurodegeneration observed in AD [23,24]. Aβ has been found in blood at nanomolar concentrations and is abundantly produced by platelets [25]. RBCs encounter Aβ peptides at the luminal surface level of brain capillaries [26] and seem to only interact with monomeric Aβ peptides [27]. Aβ alters RBC metabolism and ATP release and induces RBC death [28,29,30,31] through a signaling pathway involving, among other factors, caspase 3 and protein kinase C [31,32,33,34,35,36]. Evidence from epidemiological data indicates a close association between vascular and AD pathology [37]. However, experimental studies suggest that Aβ can reduce cerebral blood flow (CBF), inducing neurovascular dysfunction and increasing the brain’s susceptibility to ischemia [38]. Therefore, we are interested in determining whether RBCs contribute to AD pathogenesis. Previous studies have reported decreased S1P levels in AD tissues and plasma [39,40]. S1P protects neuronal cells from apoptosis [41], notably in response to Aβ [42]. Moreover, a recent paper demonstrated that S1P abrogates the neuronal Ca^2+^ dyshomeostasis induced by toxic Aβ cells [43].

Based on the importance of vascular dysfunction in AD pathology, in this study, we investigated the protective role of S1P against Aβ peptides in ATP release in RBCs.

## 2. Results

### 2.1. Protective Role of Sphingosine-1-phosphate on ATP Release

It is known that RBCs can readily uptake exogenous S1P, up to 5 μmol L^−1^ in an in vitro system [3]. Firstly, we assessed whether S1P affected the mechanism responsible for ATP release from RBCs. Here, RBCs were treated at high and low oxygen tensions with S1P at concentrations of 0.1 and 0.5 μM for 24 h. The ATP values were significantly higher for control cells with S1P at 0.5 μM compared to 0.1 μM (Figure 1A). When 0.1 μM Aβ was added to RBCs at low and high oxygen tensions for 24 h, it inhibited the release of ATP from RBCs at a low oxygen tension (Figure 1B), as previously reported [31]. Next, to verify the protective role of S1P against Aβ, S1P was pre-incubated with RBCs for 30 min before Aβ exposure at a low oxygen tension. As shown in Figure 1B, the ATP values were fully restored in the presence of 0.5 μM of S1P, with a slight protective effect at 0.1 μM. It is known that caspase-3 is involved in the mechanism responsible for the inhibition of ATP release from RBCs by Aβ [31]. Next, we examined whether the protective effect of S1P against Aβ was mediated by caspase 3. The pre-treatment of RBCs exposed to Aβ with a caspase-3 inhibitor, i.e., Z-DEVD-FMK, rescued ATP levels back to control levels (Figure 1B). In RBCs, it has been shown that ATP release is linked to a pathway involving Gi and adenylyl cyclase (AC) [18]. Mastoparan 7 (mas 7), an activator of Gi, was used to clarify the involvement of the Gi-related pathway in the protective role of S1P against Aβ. As reported in Figure 2, in the experiments with mas 7, the ATP release values remained similar between RBCs in the presence and absence of S1P, demonstrating that Gi proteins do not mediate S1P action.

### 2.2. Effect of Sphingosine-1-phosphate on the Accumulation of cAMP

Then, we investigated whether cAMP was involved in the protective effect of S1P against Aβ in deoxygenated RBCs. Here, in deoxygenated RBCs treated for 24 h with S1P alone at 0.1 and 0.5 μM, the cAMP values were significantly higher for control cells treated with S1P at 0.5 μM, with no effects observed at 0.1 μM (Figure 3). Next, to further verify the protective role of S1P against Aβ, S1P was pre-incubated with RBCs for 30 min before Aβ exposure at a low oxygen tension. As shown in Figure 3, the cAMP values were restored with 0.1 and 0.5 μM of S1P. Furthermore, the role of caspase 3 in the S1P-related mechanism was investigated; the pre-treatment of Aβ-exposed RBCs with a caspase-3 inhibitor, i.e., Z-DEVD-FMK, rescued cAMP levels to those shown by control cells.

### 2.3. Effect of Sphingosine-1-phosphate on 2,3 DPG Levels

When RBCs were treated with Aβ for 24 h, the 2,3 DPG levels observed in the deoxygenated RBCs were significantly reduced compared to the control cells (Figure 4). S1P alone at 0.5 μM increased the 2,3 DPG levels compared to the control, demonstrating that S1P could increase metabolic fluxes through glycolysis to generate 2,3-BPG, as previously reported [16]. In RBCs pre-incubated for 30 min with S1P at 0.5 μM before Aβ, the 2,3 DPG levels were significantly higher than those shown by the Aβ-treated cells. Furthermore, the role of caspase 3 in the S1P-related mechanism was studied; the pre-treatment of Aβ-exposed RBCs with Z-DEVD-FMK restored the 2,3 DPG levels, similar to S1P.

### 2.4. Effect of Sphingosine-1-phosphate on Caspase-3 Activity

Band 3 degradation by caspase-3 has been suggested to induce cdb3/deoxyHb binding site disruption in RBCs [29,30]. Cdb3/deoxyHb binding activated the pathway responsible for ATP release from deoxygenated RBCs [18]. Aβ inhibits ATP release from RBCs through a pathway involving the activation of caspase-3 [31]. As shown in Figure 5, Aβ treatment dramatically increased caspase-3 activity in a time-dependent manner. Aβ-mediated caspase-3 activation was significantly rescued by pre-incubation with S1P at 0.5 μM for 30 min, with only a minor protective effect observed with 0.1 μM. The pre-incubation of Aβ-treated RBCs with Z-DEVD-FMK inhibited the Aβ-mediated caspase-3 activation. However, this observation excluded the presence of unspecified proteolytic activities. Moreover, S1P alone at 0.5 μM did not affect caspase-3 activity. Then, we examined the effects of mas 7, an activator of Gi, to determine whether Gi mediated the observed protective effect of S1P against the activation of caspase-3 by Aβ. As reported in Figure 5, caspase-3 was unaffected in the presence of mastoparan 7, demonstrating that Gi proteins do not mediate S1P action.

### 2.5. Hemolysis Degree

The spontaneous lysis of RBCs is another potential source of extracellular ATP. Thus, the RBC suspensions were analyzed to evaluate the hemoglobin concentrations in the supernatants and determine hemolysis after the experiments [40]. In all experiments, hemolysis was less than ~3%.

## 3. Discussion

RBCs release ATP in response to low oxygen tension [18]. The starting event in the release of ATP from RBCs involves an interaction between deoxyHb and the cytoplasmic domain of the anion exchange protein band 3, i.e., the cdb3–deoxyHb/band 3 complex induces stress in the membrane components, triggering the downstream pathway responsible for ATP release. It has been shown that ATP release and cAMP accumulation are strongly reduced in RBCs in the presence of Aβ and are associated with caspase-3 activation [31], thus decreasing tissue oxygenation, particularly in cerebral microvascular circulation, and aggravating AD pathology. Here, we report that the Aβ-mediated inhibition of ATP release from deoxygenated RBCs was abolished when cells were pre-incubated with sphingosine-1-phosphate (S1P) before treatment with Aβ. The signalling pathway underlying ATP release from RBCs includes the heterotrimeric G proteins Gs and Gi, adenylyl cyclase (AC), and cyclic AMP-dependent protein kinase A [18]. In the presence of S1P, comparable amounts of intracellular cAMP were measured following incubation with mas 7 (i.e., stimulatory agent of Gi), both in the presence and absence of Aβ peptides; this finding suggests that the activity of the Gi subunit in heterotrimeric G proteins could not explain the protective effect induced by S1P in Aβ-treated RBCs. The possible role of S1P in AD is controversial, with some studies suggesting a causative role in AD and others proposing a protective role [44]. We observed that pre-treatment with a caspase-3 inhibitor, i.e., Z-DEVD-FMK, before Aβ rescued the ATP and cAMP levels to those observed in the control cells, similar to that shown by S1P. On these bases, we suggest that S1P inhibited the Aβ-mediated activation of caspase-3 activity, protecting the cytoplasmic domain of the anion exchange protein band 3, i.e., cdb3, through caspase-3 cleavage.

Since the release of ATP from RBCs occurs in response to low oxygen tension and consists of an interaction between deoxyHb and cdb3 [18], our findings suggest that the mechanism underlying the protective role of S1P in the inhibition of ATP release, triggered by Aβ, partially involves the S1P-mediated abrogation of caspase-3 activation. These findings align with a previous paper, which showed that the S1P agonist SEW2871 decreased Aβ-induced caspase-3 activation, neuronal death, and cognitive damage in rats with AD [45]. Furthermore, we showed that S1P increased the 2,3-DPG levels within the cell. In a previous study [16], it has been suggested that S1P induces 2,3 DPG production by binding directly to deoxy-Hb, thereby stabilizing Hb in the deoxygenated state. DeoxyHb binds to cdb3, triggering the release of some glycolytic enzymes to the cytosol, thereby increasing glycolysis flux to produce more 2,3-DPG. Thus, the increase in 2,3 DPG can bind more oxyHb molecules, meaning that S1P promotes the anchoring of deoxyHb to cdb3 and triggers the mechanism responsible for ATP release from RBCs in response to low oxygen tension.

## 4. Materials and Methods

### 4.1. Chemicals

Ab peptide (1–42) with a purity of >98% was purchased from Peptide Speciality Laboratories GmbH (Heidelberg, Germany). Peptides were solubilized in 100% 1,1,1,3,3,3-hexafluoro-2-propanol (HFIP; Sigma, St. Louis, MO, USA). The HFIP was then removed by vacuum evaporation, and the remaining disaggregated peptide was dissolved in dimethylsulphoxide (DMSO). Sphingosine-1-phosphate (S1P) and other chemicals were purchased from Sigma Aldrich (St. Louis, MO, USA).

### 4.2. Preparation of Red Blood Cells and Incubation Conditions

After the receipt of written informed consent from healthy volunteers, blood was voluntarily donated for the sole purpose of this study in accordance with the Declaration of Helsinki. Blood samples were collected in citrate and washed three times with an iso-osmotic NaCl solution. Low-speed centrifugation (800× *g*, 5 min) was performed to separate the plasma, avoiding mechanical stress that could cause RBC morphological alterations. Ficoll was used to isolate mature RBCs for a density gradient centrifugation. RBCs were incubated at 37 °C for 24 h with or without 0.1 μM of Aβ peptide, pre-incubated in the presence and absence of S1P at 0.1 and 0.5 μM. In experiments performed under low-oxygen conditions, the measured percentage of deoxyHb was 60 ± 0.32%. RBCs were sedimented by centrifugation at 500× *g* for 10 min to exclude the possibility that RBC lysis affected our determinations. Oxygenated hemoglobin in the supernatant was determined by light absorption at 405 nm (Cary 3E, Varian, Palo Alto, CA, USA) [46]. Although this method does not measure methemoglobin and oxidized forms of hemoglobin (about 1–3% of the total hemoglobin), it is commonly used when measuring experimentally induced RBC lysis [47].

### 4.3. ATP Assay

The luciferin–luciferase technique was used to measure ATP, as reported in [48]; this uses the ATP concentration dependence of light generated by the reaction of ATP with firefly tail extract.

### 4.4. Measurement of cAMP

After RBC exposure to different experimental conditions, they were added to 4 mL of ice-cold absolute ethanol containing HCl (1 mmol/L), and the mixtures were centrifuged at 14,000× *g* for 10 min at 4 °C. The supernatants were removed and stored overnight at −20 °C to precipitate the remaining proteins. The samples were then centrifuged a second time at 3700× *g* for 10 min at 4 °C. The supernatant was removed and dried under vacuum centrifugation. cAMP’s concentration was then determined, as previously described [49], with a cAMP Biotrak enzyme immunoassay system (Amersham Biosciences, Amersham, UK).

### 4.5. Determination of 2,3 DPG

2,3-DPG in 20 μL of RBC pellet was isolated with 100 μL of 0.6 M cold perchloric acid on ice, vortexed, and centrifuged. The homogenate was centrifuged at 20,000× *g* for 10 min. A volume of 80 μL supernatant was transferred to a new tube, neutralized with 10 μL of 2.5 M K_2_CO_3_, and centrifuged. An aliquot of supernatant was used to measure the 2,3-DPG levels using a commercially available kit (Sigma Aldrich, St. Louise, MO, USA).

### 4.6. Caspase-3 Activity Determination

After RBC exposure to different experimental conditions, the caspase activity was evaluated, as previously described [50]. The DEVD-dependent protease activity was determined using N-Acetyl-Asp-Glu-Val-Asp p-nitroanilide as a substrate and the immunosorbent caspase-3 activity assay kit from Roche Molecular Biochemicals. Briefly, after different RBC lysates treatments were prepared, caspase 3 was captured from the lysate in microplates coated with anti-caspase 3 monoclonal antibodies. After washing the plates, N-Acetyl-Asp-Glu-Val-Asp p-nitroanilide was added, and the released free p-nitroanilide (pNA) was determined spectrophotometrically at 405 nm. A pNA calibration curve was plotted from a pNA stock solution, and the caspase-3 activity was measured relative to this curve.

### 4.7. Statistical Analysis

All data are expressed as means ± SD. Statistical analyses (Student’s test and ANOVA) were performed with SYSTAT 10.2 software (Statcom, Inc., Richmond, CA, USA). The level of significance was set at 0.05. Inter-assay (as an estimation of the reproducibility) precisions were shown in Appendix A (Appendix A).

## 5. Conclusions

We prove that S1P rescued the inhibition of ATP release from RBCs triggered by Ab. Among the several signalling pathways mediated by S1P, our results suggest that the protective path involves caspase-3 inhibition. The protective role of S1P could be relevant to supporting the energy demands in tissues, particularly in cerebral microvascular regions after ischemia or where a deposition may cause the cerebral vessel lumen to narrow. While this is a promising finding, this study is limited because it did not use AD models; therefore, future studies that use blood cells from AD patients are warranted. Data from this study indicate that S1P is a possible agent for treating or preventing AD.

## Figures and Tables

**Figure 1 ijms-25-05184-f001:**
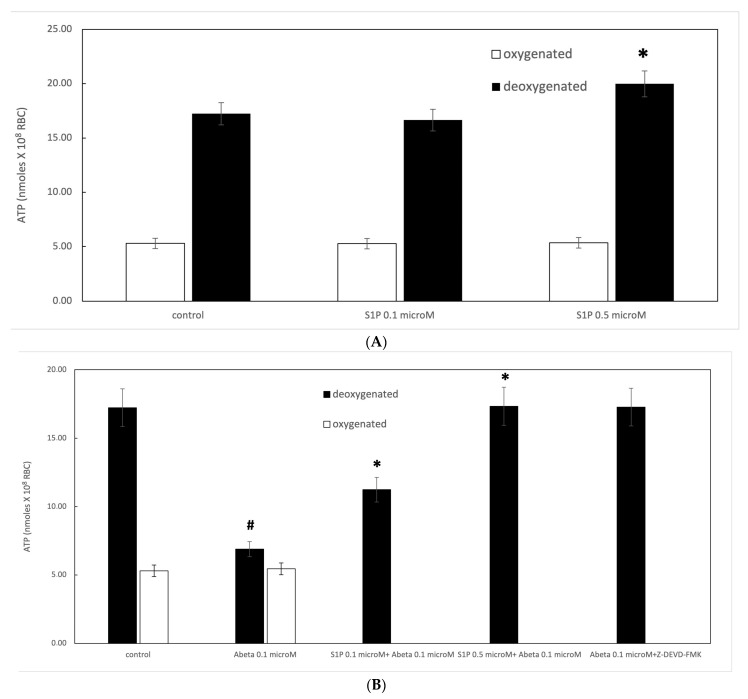
(**A**) Effect of sphingosine-1-phosphate (S1P) on ATP release in oxygenated (white) and deoxygenated (black) red blood cells (RBCs). Values are presented as the mean ± SD (*N* = 5). * *p* < 0.05 compared with control. (**B**) Protective role of S1P against amyloid beta (Aβ) peptides. Values are presented as the mean ± SD (*N* = 5). # *p* < 0.05 compared with deoxygenated cells, * *p* < 0.05 compared with Aβ.

**Figure 2 ijms-25-05184-f002:**
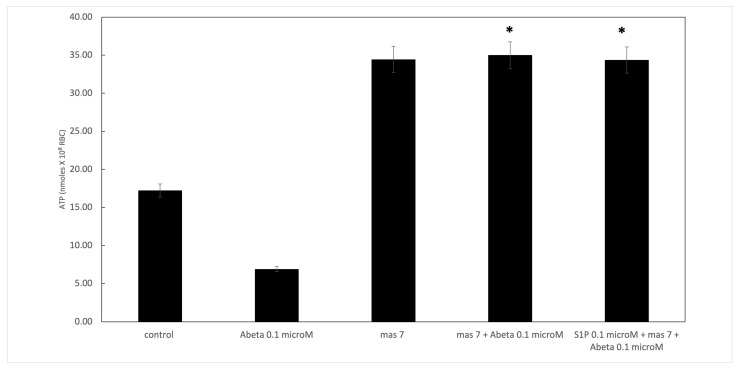
Effect of mastoparan 7 (mas 7) at 10 µM on ATP release from RBCs. Values are presented as the mean ± SD (*N* = 6). * *p* < 0.05 compared with Aβ cells.

**Figure 3 ijms-25-05184-f003:**
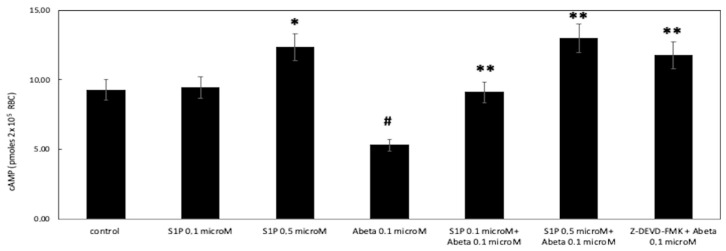
Effect of S1P on cyclic adenosine monophosphate (cAMP) levels in Ab-treated and un-treated deoxygenated RBCs (black). Values are presented as the mean ± SD (*N* = 5). * *p* < 0.05 compared with control. # *p* < 0.05 compared with deoxygenated cells, ** *p* < 0.05 compared with Aβ.

**Figure 4 ijms-25-05184-f004:**
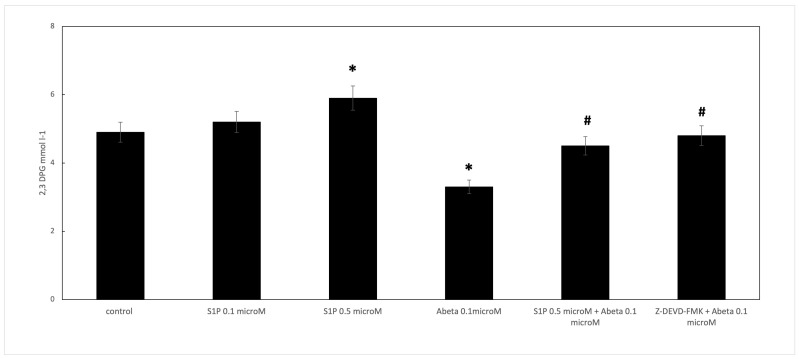
Effect of S1P treatment on 2,3-diphosphoglycerate (2,3 DPG) levels in deoxygenated RBCs. Values are presented as the mean ± SD (*N* = 5). * *p* < 0.01 compared with control deoxygenated cells. # *p* < 0.01 compared with Aβ-treated cells.

**Figure 5 ijms-25-05184-f005:**
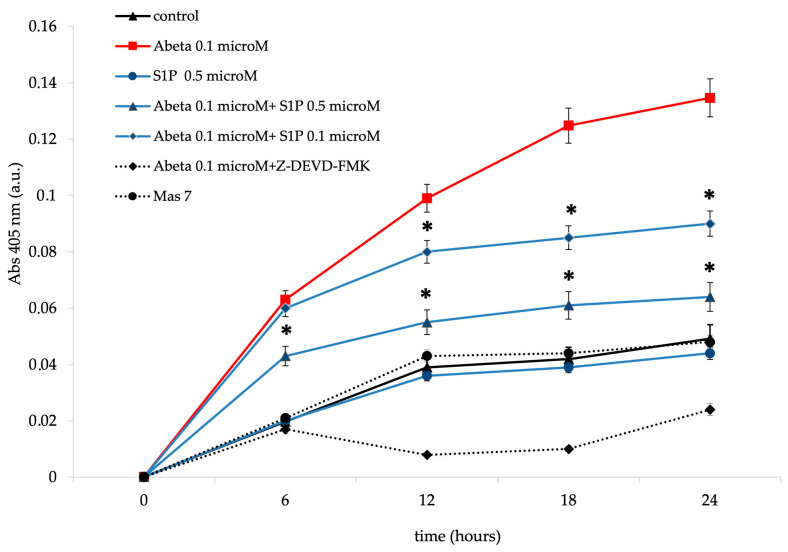
Caspase-3 activity in deoxygenated RBCs following treatment under different conditions. Values are presented as the mean ± SD (*N* = 5). * *p* < 0.05 compared with Aβ-treated cells. a.u. (Absorption units).

## Data Availability

All data points generated or analyzed during this study are included in this article, and no further underlying data are necessary to reproduce the results.

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
