# Peer review of "Sphingosine-1-phosphate Decreases Erythrocyte Dysfunction Induced by β-Amyloid"

_ijms, 2024, doi:10.3390/ijms25105184_

Round 1

Reviewer 1 Report (New Reviewer)

Comments and Suggestions for Authors

In their manuscript, Misiti et al. provide data regarding the protective role of sphingosine-1-phosphate on RBCs that simulate the ones of Alzheimer's disease patients. Their findings are original and of importance to better understand the molecular basis of these interactions/effects. Moreover, it is a good fit for the selected special issue and will be of interest to the readership of the journal.

Nonetheless, there are some important points to be checked before its acceptance.

1) In the Introduction section the authors should mention the role of caspase-3 in ATP release inhibition (this information is currently first mentioned in the results) so that the reader can have a better background and more easily understand the workflow of the paper.

2) In the Results section, lines 80-83 the authors mention that " Pre-treatment of RBCs exposed to Ab with a caspase-3 inhibitor, i.e., Z-DEVD-FMK, was able to rescue ATP levels back to control levels (Fig. 1B), evidencing the involvement of caspase-3 in the protective mechanism elicited by S1P. I have two concerns regarding this sentence.

 - My major concern is how the authors concluded that caspase-3 plays a role in the protective mechanism elicited by S1P, since S1P was not added in the sample with the inhibitor. This result clearly shows that caspase-3 plays a role in Ab-mediated inhibition of ATP release, but the fact that S1P implicates caspase-3 in its function is not supported by these results and is mainly an, albeit logical, hypothesis. It would be more solid if it was discussed later on, in the discussion section, by combining the results of ATP release (Figure 1B) and caspase-3 activity (Figure 5), since S1P leads to a minor caspase-3 activation.

- My other concern is mostly syntax-related. The fact that it is written that "caspase-3 plays a role in the protective mechanism.." makes the reader get the impression that caspase-3 acts beneficially. So, it would be preferable to alter this sentence to make it clearer.

3) I have the same concerns when reading the lines 123-125, especially since I did not find the effect of the inhibitor in the Figure that concerns cAMP.

4) The same is true for lines 137-139 and the implication of the inhibitor in 2,3-DPG restoration.

5) In Figure 1 it would be helpful to add the word "release" on y axis so that it is clear to the reader that the Figure shows ATP release and not content.

6) Finally, could the authors add a little more information regarding the Casp-3 measurement assay? 

Comments on the Quality of English Language

There are some not so clear sentences, in terms of syntax, throughout the manuscript, therefore an extensive proofreading would be beneficial.

Author Response

Dear Reviewers,

On behalf of all the authors, I sincerely thank you for your letter and the reviewers for their constructive comments on our article. These comments are valuable and contribute to the quality of our article.

Based on comments from reviewers and our self-examination, we revised the manuscript extensively. In this revision, revised sections of the manuscript are marked in yellow.

I appreciate your consideration. I look forward to hearing from you.

Reviewer 1

In their manuscript, Misiti et al. provide data regarding the protective role of sphingosine-1-phosphate on RBCs that simulate the ones of Alzheimer's disease patients. Their findings are original and of importance to better understand the molecular basis of these interactions/effects. Moreover, it is a good fit for the selected special issue and will be of interest to the readership of the journal.

Nonetheless, there are some important points to be checked before its acceptance.

In the Introduction section the authors should mention the role of caspase-3 in ATP release inhibition (this information is currently first mentioned in the results) so that the reader can have a better background and more easily understand the workflow of the paper.

Thank you for your review. Your review will help to improve the quality of our article. In the introduction section of the revised version, We have mentioned the role of caspase in the ATP release mechanism in RBC.

In the Results section, lines 80-83 the authors mention that " Pre-treatment of RBCs exposed to Ab with a caspase-3 inhibitor, i.e., Z-DEVD-FMK, was able to rescue ATP levels back to control levels (Fig. 1B), evidencing the involvement of caspase-3 in the protective mechanism elicited by S1P. I have two concerns regarding this sentence.

 - My major concern is how the authors concluded that caspase-3 plays a role in the protective mechanism elicited by S1P, since S1P was not added in the sample with the inhibitor. This result clearly shows that caspase-3 plays a role in Ab-mediated inhibition of ATP release, but the fact that S1P implicates caspase-3 in its function is not supported by these results and is mainly an, albeit logical, hypothesis. It would be more solid if it was discussed later on, in the discussion section, by combining the results of ATP release (Figure 1B) and caspase-3 activity (Figure 5), since S1P leads to a minor caspase-3 activation.

- My other concern is mostly syntax-related. The fact that it is written that "caspase-3 plays a role in the protective mechanism.." makes the reader get the impression that caspase-3 acts beneficially. So, it would be preferable to alter this sentence to make it clearer.

Thank you for your review. As suggested, we have moved the discussion on the role of caspase to the discussion section. At the same time, in the revised version, we have rewritten some sentences in the manuscript to avoid misunderstanding on the role of capsase 3.

I have the same concerns when reading the lines 123-125, especially since I did not find the effect of the inhibitor in the Figure that concerns cAMP.

Thank you for your review. As suggested, we have added data to the caspase 3 inhibitor in Figure 3; it has been our negligence, so we have moved the discussion on the role of caspase to the discussion section.

The same is true for lines 137-139 and the implication of the inhibitor in 2,3-DPG restoration.

Thank you for your review. As suggested previously for Figures 1 and 3, also for Figure 4, we have moved the discussion on the role of caspase to the discussion section.

In Figure 1 it would be helpful to add the word "release" on y axis so that it is clear to the reader that the Figure shows ATP release and not content.

Thank you very much for your valuable advice. We have corrected this in Figure 1 in the revised manuscript. 

Finally, could the authors add a little more information regarding the Casp-3 measurement assay? 

Thanks for your suggestions. The materials and methods section has been improved in the revised manuscript,

Reviewer 2 Report (New Reviewer)

Comments and Suggestions for Authors

The participation of beta-amyloid in the functioning of various body systems still remains poorly understood. This also applies to blood cells. This is an interesting article that describes the role of sphingosine-1-phosphate (S1P) in signaling pathways involved in the mechanism underlying ATP release in Aβ-treated RBCs. The article is written in a rather lapidary style and some points are missed. Major points:

1. It is necessary to combine the data in Figure 3 into one panel for easy comparison.

2. Indicate the concentration of beta-amyloid in the figure captions.

3. Lines 118-119 indicate that “cAMP levels in RBCs are significantly

higher when the cells are deoxygenated compared to oxygenated conditions (Fig. 3A)." However, Figure 3A shows data only for oxygenated conditions.

4. There is no data for caspase 3 inhibitor in Figure 3.

5. The source of RBC is not indicated in the materials and methods.

6. Materials and methods are written very briefly and need to be described in more detail.

Author Response

Dear Reviewers,

On behalf of all the authors, I sincerely thank you for your letter and the reviewers for their constructive comments on our article. These comments are valuable and contribute to the quality of our article.

Based on comments from reviewers and our self-examination, we revised the manuscript extensively. In this revision, revised sections of the manuscript are marked in yellow.

I appreciate your consideration. I look forward to hearing from you.

Reviewer 2

The participation of beta-amyloid in the functioning of various body systems still remains poorly understood. This also applies to blood cells. This is an interesting article that describes the role of sphingosine-1-phosphate (S1P) in signaling pathways involved in the mechanism underlying ATP release in Aβ-treated RBCs. The article is written in a rather lapidary style and some points are missed. Major points:

It is necessary to combine the data in Figure 3 into one panel for easy comparison.

Thank you for your review. Your review will help to improve the quality of our article. In the revised version, Figures 3A and 3B have been unified in one panel (Figure 3) to better compare experimental conditions.

Indicate the concentration of beta-amyloid in the figure captions.

Thank you for your review. It is our negligence that we did not show the beta concentration in the figure caption. The concentration of beta has been added in the Figures of the revised version.

Lines 118-119 indicate that “cAMP levels in RBCs are significantly higher when the cells are deoxygenated compared to oxygenated conditions (Fig. 3A)." However, Figure 3A shows data only for oxygenated conditions.

Thank you for your review. This was our mistake, and we have revised it in the manuscript.

There is no data for caspase 3 inhibitor in Figure 3.

Thank you for your review. It has been our negligence. As suggested, we have added data to the caspase 3 inhibitor in Figure 3 in the revised version.

The source of RBC is not indicated in the materials and methods.

Thank you for your review. It has been our negligence. As suggested, we have added missing data in the revised manuscript,

Materials and methods are written very briefly and need to be described in more detail.

Thanks for your suggestions. The materials and methods section has been improved in the revised manuscript.

Round 2

Reviewer 1 Report (New Reviewer)

Comments and Suggestions for Authors

The authors adequately addressed all my concerns, therefore I believe the work can be accepted for publication in its current form.

This manuscript is a resubmission of an earlier submission. The following is a list of the peer review reports and author responses from that submission.

Round 1

Reviewer 1 Report

Comments and Suggestions for Authors

In this study the authors evaluated the role of S1P on the release of ATP after Ab peptide treatment of red blood cells.

The current manuscript has several weaknesses and must be definetively improved. The introduction does not provide sufficient information to understand the scope of the experimental plan; materials and methods need to be detailed; the results are described in a superfical manner; the conclusions are not supported by the evidence here presented.  

The simple treatment of red blood cells (of unknown origin) with Aβ peptide is not sufficient to reproduce Alzheimer's disease or in any case it is not sufficient to draw conclusions on the neuroprotective mechanism of S1P in Alzheimer's disease, as stated in both the abstract and discussion section. 

Author Response

Response to Reviewer 1 Comments

Comment 1

The current manuscript has several weaknesses and must be definitively improved. The introduction does not provide sufficient information to understand the scope of the experimental plan; materials and methods need to be detailed; the results are described superficially; the conclusions are not supported by the evidence presented here.  

Response 1.

Thank you for the comment. According to the suggestion, we have deeply revised the Abstract, Introduction and Discussion sections to define the main aim of our paper better. The material and Methods section has been improved with new references; all methods reported show references or the assay kit company’s description. Regarding the Results section, we have taken into consideration the suggestion of the reviewer; we have corrected abbreviations and changed the legend’s colour to make it easy to read, as well as the text revision of the entire manuscript.

Comment 2

The simple treatment of red blood cells (of unknown origin) with Aβ peptide is insufficient to reproduce Alzheimer's disease. In any case, it is insufficient to conclude the neuroprotective mechanism of S1P in Alzheimer's disease, as stated in the abstract and discussion section. 

Response 2. Thank you for the comment. Discussion and conclusion have been revised in the text to focus on S1P's protective role against Abeta toxicity in RBCs instead of AD pathology. Based on the largely known studies that show a relationship between Abeta peptide and AD, and with the support of a recent study [reference 47] regarding the beneficial role of S1P in mouse models of AD, in briefly we conclude our manuscript suggesting that our research on RBCs might open new therapies in AD.

Reviewer 2 Report

Comments and Suggestions for Authors

Thank you for your work!

The communication contains important information about the functional consequences of S1P-mediated effects in Alzheimer's disease RBCs. After minor corrections, the communication can be published.

The comments include requirements and recommendations.

1) The abbreviations ‘AD’ (line 11) and ‘’ (line 12), ‘DPG’ (line 32), ‘PKC’ (line 42), ‘PO2’ (line 83), ‘DEVD’ (line 104), ‘pNA’ (line 104) are not deciphered (at first mention) or don't use them in the abbreviation.

2) Introduction (line 21):

There is no information about Alzheimer's disease, perhaps it is worth briefly writing something about it in one sentence or phrase. In the list of keywords you mention ‘Band 3’ (line 18), in the introduction you don't say anything about it, briefly explain what it is. ‘Abeta’ (line 40): Introduce uniform terminology for all text and figures.

3) ‘… the Ethical Committee of our University…’ (line 63): It will be correct to indicate which university

4) ‘CAMPʹs concentration’ (line 96): If you meant cAMP, then correct the register

5) Figure 1 (line 134):

Move the figure to the part of the text where you first mentioned it (line 122). It is difficult to perceive the text in conjunction with the figure. Why don't you refer to Figure 1A (line 134) and Figure 1B (line 136)? Perhaps this would make it easier to understand. Figure 1A and Figure 1B should be one figure, but you have 2. ‘…Aβ-treated cells…’ (line 122) and ‘abeta’: make sure that the terminology in the text, figure and captions match. Reduce the size of the graphs, it is very difficult to read the caption for the figure. What is 'microM'? There is no mention in the text or in the figure caption.

5) Figure 2 (line 145):

As shown in Figure 2, in the presence 130 of mastoparan 7, ATP release values…’ (lines 130-131): Most likely you incorrectly referred, the figure shows cAMP accumulation, you need to fix it.

5) Figure 3 (line 175):

Figure 3A and Figure 3B should be one figure, but you have 2. Reduce the size of the graphs, figure and caption must be on the same page. There is a place for labeling curves in the Figure 3A, this will help to facilitate perception. ‘Abs 405 nm (a.u.)’: nowhere is it explained what it is, the abbreviation is not deciphered, explain it in the figure caption. Make sure that the terminology in the text, figure and captions match.

6) ‘The percentage of hemolysis was always less 190 than 3%’ (lines 190-191): How was it fixed? What indicators? If you cannot give exact data and statistics, explain why.

7) References (line 239):

36. Sprague, R.S.; Ellsworth, M.L.; Stephenson, A.H.; Lonigro, A.J. Participation of cAMP in a signal-transduction pathway relating erythrocyte deformation to ATP release…’ (lines 318-319): It is not very clear what exactly was borrowed in the methods from this article. Please specify which approach was used.

RBCs have been shown to regulate vascular resistance in the lung [15]…’ (line 193): This article is about skeletal muscles, correct the link or thesis.

The References needs to be corrected: Starting from line 269, there is a duplication of reference numbers; starting from line 278, numbering is lost.

Author Response

Response to Reviewer 2 Comments

Comment 1

1) The abbreviations ‘AD’ (line 11) and ‘’ (line 12), ‘DPG’ (line 32), ‘PKC’ (line 42), ‘PO2’ (line 83), ‘DEVD’ (line 104), ‘pNA’ (line 104) are not deciphered (at first mention) or don't use them in the abbreviation.

Response 1. Thank you for the comment. All abbreviations reported have been deciphered.

 Comment 2

2) Introduction (line 21):

There is no information about Alzheimer's disease, perhaps it is worth briefly writing something about it in one sentence or phrase. In the list of keywords, you mention ‘Band 3’ (line 18), in the introduction you don't say anything about it; briefly explain what it is. ‘Abeta’ (line 40): Introduce uniform terminology for all text and figures.

Response 2. Thank you for the comment, Introduction section has been deeply revised, and more information regarding Alzheimer's Disease has been inserted (with new references); As suggested by the reviewer, the term band 3 has been briefly explained; because band 3, together with “vascular disease” term do not represent the core of this paper, we have decided to cancel them from the keywords list. All abbreviations in the text have been verified and uniformed.

 Comment 3

3) ‘… the Ethical Committee of our University…’ (line 63): It will be correct to indicate which university

Response 3. Thank you for the comment; the name of the university has been specified.

 Comment 4

4) ‘CAMPʹs concentration’ (line 96): If you meant cAMP, then correct the register

Response 4. Thank you for the comment. The right abbreviation for cAMP has been inserted.

 Comment 5

5) Figure 1 (line 134):

Move the figure to the part of the text where you first mentioned it (line 122). It is difficult to perceive the text in conjunction with the figure. Why don't you refer to Figure 1A (line 134) and Figure 1B (line 136)? This may make it easier to understand. Figure 1A and Figure 1B should be one figure, but you have 2. ‘…Aβ-treated cells…’ (line 122) and ‘abeta’: make sure that the terminology in the text, figure, and captions match. Reduce the size of the graphs, it is very difficult to read the caption for the figure. What is 'microM'? There is no mention in the text or the figure caption.

Comment 6

5) Figure 2 (line 145):

As shown in Figure 2, in the presence 130 of mastoparan 7, ATP release values…’ (lines 130-131): Most likely you incorrectly referred, the figure shows cAMP accumulation, you need to fix it.

Comment 7

5) Figure 3 (line 175):

Figure 3A and Figure 3B should be one figure, but you have 2. Reduce the size of the graphs, figures and captions must be on the same page. There is a place for labelling curves in Figure 3A, which will help facilitate perception. ‘Abs 405 nm (a.u.)’: nowhere is it explained what it is, and the abbreviation is not deciphered; explain it in the figure caption. Ensure the terminology in the text, figure, and captions match.

Response 5-7. Thank you for the comments. Regarding the Figure’s suggestions, we have re-sized them and changed the legend’s colour in all Figures to make it easy to read. Figures 1A and 1B have been improved but are not unified, as the reviewer suggested; according to the author, more than six experimental conditions in the same figure could make the figure unclear to read. Figures 3A and 3B have been unified. All abbreviations in the Figure’s legend have been verified and now match those reported in the text.

 Comment 8

6) ‘The percentage of hemolysis was always less 190 than ∼3%’ (lines 190-191): How was it fixed? What indicators? If you cannot give exact data and statistics, explain why.

Response 8. Thank you for the comment. In the material and methods section, we have inserted a reference related to the method utilized to determine the hemolysis degree.

 Comment 9

7) References (line 239):

36. Sprague, R.S.; Ellsworth, M.L.; Stephenson, A.H.; Lonigro, A.J. Participation of cAMP in a signal-transduction pathway relating erythrocyte deformation to ATP release…’ (lines 318-319): It is not very clear what exactly was borrowed in the methods from this article. Please specify which approach was used.

RBCs have been shown to regulate vascular resistance in the lung [15]…’ (line 193): This article is about skeletal muscles, correct the link or thesis.

The References need to be corrected: Starting from line 269, there is a duplication of reference numbers; starting from line 278, numbering is lost.

Response 9. Thank you for the comments. References have been verified, updated, and re-numbered; the reference related to the cAMP determination method has been changed, and the information reported in the text regarding the reference [15] has been corrected.

Reviewer 3 Report

Comments and Suggestions for Authors

Knowing that the soluble oligomers arising from the aggregation of the amyloid beta peptide (Aβ) have been identified as the main pathogenic agents in Alzheimer's disease (AD) it is very important to restore the  cAPM level and to improve the brain oxygenation thus promoting neuroprotection. The study design is quite well done. The figures very faithfully describe the effect of exogenous administration of SPHINGOSINE-1-PHOSPHATE (S1P) on the restoration of cAMP reserves with the improvement of oxygenetion. 

There are similar articles (Bigi A, Cascella R, Fani G, Bernacchioni C, Cencetti F, Bruni P, Chiti F, Donati C, Cecchi C. Sphingosine 1-phosphate attenuates neuronal dysfunction induced by amyloid-β oligomers through endocytic internalization of NMDA receptors. FEBS J. 2023). The results are clearly presented. References could be up-dated. 

Author Response

Response to Reviewer 3 Comments

Comment 1

Knowing that the soluble oligomers arising from the aggregation of the amyloid beta peptide (Aβ) have been identified as the main pathogenic agents in Alzheimer's disease (AD) it is very important to restore the cAMP level and to improve the brain oxygenation thus promoting neuroprotection. The study design is quite well done. The figures faithfully describe the effect of exogenous administration of SPHINGOSINE-1-PHOSPHATE (S1P) on the restoration of cAMP reserves with the improvement of oxygenation. There are similar articles (Bigi A, Cascella R, Fani G, Bernacchioni C, Cencetti F, Bruni P, Chiti F, Donati C, Cecchi C. Sphingosine 1-phosphate attenuates neuronal dysfunction induced by amyloid-β oligomers through endocytic internalisation of NMDA receptors. FEBS J. 2023). The results are presented. References could be updated. 

Response 1. Thank you for the comments. References have been verified, updated, and re-numbered.

Round 2

Reviewer 1 Report

Comments and Suggestions for Authors

I still believe that the conlcusion are not supported by the scientific evidence here presented. The manuscript needs to be stronlgy improved not only in the written part but especially in the experimental section. Thus, more experiments are needed to get to the described conclusions. 

Author Response

Response to Reviewer 1 Comments

Comment 1

The conclusions are still not supported by the scientific evidence presented here. The manuscript needs to be strongly improved not only in the written part but especially in the experimental section. Thus, more experiments are needed to get to the described conclusions.

Response 1.

We thank the reviewer for his comments; we have further improved the conclusion, avoiding any statement not fully supported by results; other theses reported in the conclusions have been reported because they are supported by other authors, as shown by related references (see 15,46 and 47].

Regarding the reviewer’s suggestion to improve the experimental procedures, for ATP and cAMP levels determinations, we add a new experimental condition; we evaluate the effect of caspase 3 inhibitor pre-treatment prior to Ab exposure. As shown in Figures 1A and 2, in the presence of caspase 3 inhibitor ATP and cAMP levels were rescued to control cells. This suggests that caspase 3 activity is involved in the mechanism underlying ATP inhibition by Ab , and S1P affects caspase 3, like a caspase 3 inhibitor. Furthermore, we add a new table to show 2,3 DPG levels in different experimental conditions, as shown in Table 1. Data in Table 1 show that S1P supplementation improves O2 delivery in RBCs damaged by Ab,  increasing 2,3 DPG levels.

This manuscript should be published as “Communication”; we believe that for this kind of manuscript category, obtained results and experimental procedures performed are satisfactory.